# REAC Neurobiological Modulation as a Precision Medicine Treatment for Fibromyalgia

**DOI:** 10.3390/jpm13060902

**Published:** 2023-05-27

**Authors:** Analízia Silva, Ana Rita Barcessat, Rebeca Gonçalves, Cleuton Landre, Lethícia Brandão, Lucas Nunes, Hyan Feitosa, Leonardo Costa, Raquel Silva, Emanuel de Lima, Ester Suane Monteiro, Arianna Rinaldi, Vania Fontani, Salvatore Rinaldi

**Affiliations:** 1Department of Biological and Health Sciences, Federal University of Amapá—UNIFAP, Macapá 68903-419, Brazil; analiziapena@gmail.com (A.S.);; 2Department of Biomedical Sciences, University of Sassari, 07100 Sassari, Italy; 3Department of Adaptive Neuro Psycho Physio Pathology and Neuro Psycho Physical Optimization, Rinaldi Fontani Institute, 50144 Florence, Italy; 4Department of Regenerative Medicine, Rinaldi Fontani Institute, 50144 Florence, Italy; 5Research Department, Rinaldi Fontani Foundation, 50144 Florence, Italy

**Keywords:** fibromyalgia, pain, psychopathological symptoms, endogenous bioelectric activity

## Abstract

Fibromyalgia syndrome (FS) is a disorder characterized by widespread musculoskeletal pain and psychopathological symptoms, often associated with central pain modulation failure and dysfunctional adaptive responses to environmental stress. The Radio Electric Asymmetric Conveyer (REAC) technology is a neuromodulation technology. The aim of this study was to evaluate the effects of some REAC treatments on psychomotor responses and quality of life in 37 patients with FS. Tests were conducted before and after a single session of Neuro Postural Optimization and after a cycle of 18 sessions of Neuro Psycho Physical Optimization (NPPO), using evaluation of the functional dysmetria (FD) phenomenon, Sitting and Standing (SS), Time Up and Go (TUG) tests for motor evaluation, Fibromyalgia Impact Questionnaire (FIQ) for quality of life. The data were statistically analyzed, and the results showed a statistically significant improvement in motor response and quality of life parameters, including pain, as well as reduced FD measures in all participants. The study concludes that the neurobiological balance established by the REAC therapeutic protocols NPO and NPPO improved the dysfunctional adaptive state caused by environmental and exposomal stress in FS patients, leading to an improvement in psychomotor responses and quality of life. The findings suggest that REAC treatments could be an effective approach for FS patients, reducing the excessive use of analgesic drugs and improving daily activities.

## 1. Introduction

Fibromyalgia is a chronic disorder characterized by widespread musculoskeletal pain, fatigue, and tender points, along with other symptoms such as sleep disturbances, cognitive impairments, and mood disturbances [1].

Despite extensive research, the exact cause of fibromyalgia remains unknown, and there are several theories that have been proposed to explain its underlying mechanisms [2,3,4]; among these, the main ones are the central sensitization theory (CST) [5], the neurotransmitter imbalance theory (NIT) [6], genetic and epigenetic factors (GEF) [7], dysregulation of the immune system (DIS) [8], psychosocial and behavioral factors (PBF) [9].

CST suggests that fibromyalgia is a disorder of the central nervous system [5]. According to this theory, repeated nerve stimulation, such as from physical trauma, infections, or emotional stress, can result in an amplified pain response in the brain and spinal cord. This leads to an increased sensitivity to pain signals, and even non-painful stimuli can be perceived as painful in individuals with fibromyalgia [5]. Abnormal processing of pain signals in the central nervous system may also contribute to the heightened sensitivity to other sensory stimuli, such as light, sound, and touch, which are commonly reported by fibromyalgia patients.

NIT proposes that fibromyalgia may involve imbalances in neurotransmitters [10]. Studies have shown alterations in the levels of neurotransmitters such as serotonin [11,12], dopamine [13,14], and norepinephrine [15] in fibromyalgia patients. These neurotransmitters are involved in regulating pain perception, mood, sleep, and stress responses. Imbalances in these neurotransmitters may disrupt the normal functioning of pain modulation pathways in the brain, leading to increased pain sensitivity and other fibromyalgia symptoms [10].

Another factor that is probably common to CST and NIT in the genesis of fibromyalgia pain is the aberrant functioning of the nervous system, associated with small-fiber neuropathy [16].

GEF evidence suggests that genetic and epigenetic factors may play a role in the development of fibromyalgia [7,17]. Studies have shown that fibromyalgia tends to run in families [18], and certain gene variations have been associated with an increased risk of developing fibromyalgia [19]. Epigenetic changes, which are alterations in gene function without changes in the underlying DNA sequence, may also contribute to fibromyalgia. Environmental factors, such as early life stress, trauma, and infections, can influence gene expression and contribute to the development of fibromyalgia in susceptible individuals.

DIS has also been proposed as a possible underlying mechanism in fibromyalgia. Some studies have suggested that fibromyalgia may involve an immune system dysfunction, leading to increased inflammation and abnormal immune responses. Chronic inflammation can sensitize pain receptors and contribute to the pain and other symptoms experienced by fibromyalgia patients. Additionally, immune system dysregulation may also disrupt the normal functioning of other systems in the body, such as the endocrine and nervous systems, which could contribute to the complex symptom profile of fibromyalgia.

Finally, PBF has also been implicated in the development and maintenance of fibromyalgia. Stress [20,21], depression [22], anxiety [23], and other psychological factors [24,25] can influence pain perception, sleep, and immune function, and may contribute to the onset and exacerbation of fibromyalgia symptoms. Sleep disturbances, physical inactivity, and maladaptive coping strategies can also play a role in the perpetuation of fibromyalgia symptoms.

The rationale for this study is supported by two key factors. Firstly, the alterations in endogenous bioelectrical activity (EBA) may serve as a common denominator underlying the multifaceted nature of fibromyalgia symptoms. EBA, which encompasses the electrical activity within cells, has the potential to impact various levels of physiological functioning, ranging from molecular to systemic, and can influence central nervous system function, neurotransmitter balance, genetic and epigenetic regulation, immune system dysregulation, as well as psychosocial and behavioral factors, all of which may contribute to the complex manifestation of fibromyalgia.

Secondly, the availability of the Radio Electric Asymmetric Conveyer (REAC) technology designed to effectively restore dysfunctional modifications of EBA further supports the rationale for this study.

The REAC technology and its neurobiological therapeutic treatments have been extensively studied as a means of addressing the EBA changes [26,27,28,29,30]. The aim of REAC neurobiological treatments is to reorganize EBA at various levels, with the ultimate goal of improving conditions associated with various clinical pictures, including fibromyalgia [26,27,28,29,31,32,33,34,35,36,37,38,39].

In this context, the present study was developed to investigate the effects of some REAC neurobiological treatments on the dysfunctional and epigenetic adaptive components that contribute to the pathogenesis of fibromyalgia. By gaining a deeper understanding of these mechanisms, we hope to contribute to the development of more effective treatments for this challenging disorder.

## 2. Materials and Methods

### 2.1. Study Design and Study Timeline

This was an open-label interventional study, where both the researchers and the participants are aware of the treatment being administered, meaning there is no blinding or masking of the intervention. The study followed the ICH Guideline E8 (R1) on general considerations for clinical trials [40].

### 2.2. Power Analysis

After conducting a sample power analysis using Gpower, we set the effect size value to 0.8 (indicating a large effect size), with an alpha error probability of 0.05 and a power of 0.95. Using these parameters, we determined that a total sample size of 20 subjects was required.

### 2.3. Study Timeline

T0 pre-interventional assessments: functional dysmetria (FD) analysis, timed up-and-go test, sit-to-stand and fibromyalgia impact questionnaire; T1 administration of REAC NPO treatment single session; T2 verification of the effectiveness of the administration of the REAC NPO treatment by verifying the disappearance of the FD immediately after NPO; T3 post interventional assessments—after 18 NPPO sessions, verification of the stability of the functional dysmetria correction, timed up and go test, sit-to-stand test, and fibromyalgia impact questionnaire.

### 2.4. Population

Although only 20 subjects were required to conduct the study, we opted to include all applicants who met the inclusion criteria.

The study recruited a total of 37 fibromyalgic participants using a non-probabilistic sampling method that included written and electronic dissemination of information, as well as personal outreach to associations of patients with FS and rheumatologic clinics. 

The recruitment strategy aimed to reach a diverse group of participants with varying degrees of FS severity and treatment histories. To ensure a comprehensive understanding of the disease and its impact on participants, the study sought to recruit individuals from multiple geographic regions and demographic backgrounds. Forty-five percent identified themselves as brown, 27% as white, and 18% as a black person. 81% have completed higher education. As for occupation, it was possible to observe that most work as self-employed, 68%.

Ultimately, the diverse participant pool contributed to a rich and nuanced dataset that shed light on the complex nature of FS and its management.

The composition of the group was as follows: 1 man, 54 years old, and 36 women, ranging in age from 27 to 58 years with a mean age of 38. The total mean age of the sample was 38 years.

### 2.5. Inclusion and Exclusion Criteria 

The study’s inclusion criteria required participants to be of both genders, aged 18 or older, and have a clinical diagnosis of fibromyalgia syndrome (FMS). Additionally, participants must not have engaged in physical exercise for at least three months, to avoid influences of training on physical performances, and must not have any musculoskeletal conditions such as amputations, limb length discrepancies greater than 8 cm, or any other conditions deemed by the evaluator to potentially render the performance of functional tests unfeasible, that could interfere with their ability to participate in the evaluation. Finally, all participants were required to provide informed consent before being enrolled in the study.

On the other hand, the study’s exclusion criteria were designed to exclude participants with cognitive impairment, disabling mental disorders, or those who expressed the desire to leave the study at any time for any reason. The tests were administered at baseline and at the completion of the treatment cycle.

### 2.6. Motor Assessment

In order to evaluate participants’ functional abilities, a battery of standardized assessments was administered, including functional dysmetria (FD) assessment, Timed Up-and-Go (TUG) performance [41], and Sit-to-Stand (STS) performance [42]. 

The TUG performance is evaluated on a 5-point scale ranging from 1 (normal function) to 5 (severely abnormal function) based on the observer’s perception of the individual’s risk of falling [43]. 

In healthy adults, the time taken to complete a single STS repetition typically ranges from 2 to 3 s. However, this time can increase with age and in individuals with mobility impairments or other health conditions. The “five times sit-to-stand” test involves completing five repetitions of the STS task [44] as quickly as possible, and normative values for this test range from approximately 11 to 15 s in healthy older adults.

Functional dysmetria values do not exist, as it is a neurological diagnostic technique that focuses solely on the presence or absence of the phenomenon, rather than on quantitative measurements.

These assessments were chosen based on their reliability and validity in measuring functional abilities across a range of populations and were conducted in accordance with established guidelines to ensure consistency and accuracy of results.

### 2.7. Functional Dysmetria

FD is a neurobiological condition that refers to an impaired ability to control the range, force, and direction of movements. It is most commonly associated with cerebellar dysfunction [45], which is a brain region responsible for coordinating movements and regulating motor learning [46]. Cerebellum is also critically involved in a range of non-motor functions, including cognition, language, and emotion [47,48,49].

From an adaptive and epigenetic perspective, FD may arise due to a combination of genetic and environmental factors. For example, certain genetic mutations may predispose individuals to cerebellar dysfunction [50,51,52], while environmental factors such as stress or traumatic experiences may exacerbate this dysfunction and lead to further motor [53] and non-motor problems [3,54]. The concept of FD is clinically significant because FD can be a semiological sign of the presence of dysfunctional adaptive neurobiological and epigenetic modifications [46]. 

The assessment of FD involves the observation of a dysfunctional adaptive motor behavior, rather than a morphological modification. During the motor task of the assessment maneuver, FD is not perceptible to the subject but evident to the examiner [27,29,46]. To evaluate the presence of FD, participants were instructed to lie down on a medical examination table and move from a supine to a sitting position [27,29,46]. The examiner placed their hands lightly on the femoral quadriceps of the subject to perceive muscle contraction and movement without opposing it, ensuring the alignment of their left and right thumbs. During the execution of this motor task, examiners observed a progressive misalignment of the two thumbs, which was measured using a tape measure or decimeter. This motor task allows the examiner to detect the asymmetric activation (dysmetria) of symmetrical muscle groups, such as the quadriceps muscles.

### 2.8. Timed Up and Go Test

The Timed Up and Go (TUG) test [41] is a commonly utilized measure of mobility and fall risk among older adults [55]. The test was conducted in accordance with established protocols, which involved participants standing up from a chair, walking a distance of three meters, turning around, walking back to the chair, and sitting down. The TUG test has been validated and widely employed in clinical settings to evaluate balance and mobility in older adults, with previous research indicating that longer TUG times are associated with increased fall risk and reduced functional performance. The TUG test is considered a reliable and valid tool for assessing mobility and fall risk and has been recommended by several clinical practice guidelines for adult populations [55,56,57,58].

### 2.9. Sit-To-Stand Test

The Sit-to-Stand (STS) test [42] was utilized as an assessment tool for evaluating lower limb strength and power in the study participants. The STS test was performed using both the Sit-to-Stand (SS) and Stand-to-Sit (STS) variations, in which participants were instructed to rapidly rise from a seated position and return to sitting for a set number of repetitions. The SS test measures the time taken to rise from a chair to a fully standing position, while the STS test measures the time taken to sit down again after standing up. These tests have been widely used in clinical and research settings as an objective measure of lower limb strength and power and are considered to be a reliable indicator of functional capacity in older adults and individuals with musculoskeletal disorders [59,60,61,62]. By utilizing these tests in our study, we were able to obtain quantitative data on the participants’ lower limb strength and power, which was used to evaluate the efficacy of the intervention under investigation.

### 2.10. Quality of Life Assessment—Fibromyalgia Impact Questionnaire

The Fibromyalgia Impact Questionnaire (FIQ) is a self-reported outcome measure designed to assess the health status and functional disability of individuals with fibromyalgia. It was developed in the late 1980s by Burckhardt and colleagues [63] and has since been widely used in clinical and research settings [64,65].

The FIQ consists of 10 items that assess various domains of functional impairment and symptom severity over the past week, including physical functioning, work status, depression, anxiety, pain, stiffness, fatigue, sleep disturbance, and overall well-being. Each item is scored on a numerical rating scale ranging from 0 to 10, with higher scores indicating greater severity of symptoms or disability.

The physical functioning item assesses the degree to which fibromyalgia interferes with activities such as bending, lifting, and walking. Work status assesses whether the individual is able to work and whether fibromyalgia interferes with work-related tasks. Depression and anxiety items assess the degree to which these conditions are present and impact the individual’s daily life. Pain, stiffness, and fatigue items assess the severity of these symptoms over the past week. Sleep disturbance assesses the frequency and severity of sleep problems over the past week. The final item, overall well-being, asks the individual to rate their overall sense of well-being on a scale from 0 (very poor) to 10 (very good).

The FIQ has been found to have good reliability and validity in assessing the impact of fibromyalgia on individuals’ lives and has been used in numerous clinical trials and observational studies [66,67]. It is a widely recognized and validated tool for measuring the impact of fibromyalgia on patients’ health-related quality of life and functional status. The FIQ was administered in both pre-treatment and post-treatment cycles. 

### 2.11. REAC Technology

Radio Electric Asymmetric Conveyer (REAC) technology is a non-invasive, neuro-biological modulation approach that modulates cellular function through the emission of specific radioelectric signals conveyed asymmetrically inside the body. These signals interact with the cellular endogenous bioelectric field, leading to improved cellular communication, metabolic activity, and tissue repair. The underlying mechanism of REAC technology involves the generation of an asymmetrical radioelectric field, which induces a potential difference across the cell membrane, resulting in increased cellular activity. This field also triggers the flow of ions through the cell membrane, further enhancing the biological effects through interactions with the endogenous bioelectric field. The REAC device (BENE 110, ASMED Srl, Scandicci Florence, Italy) is capable of producing customized radioelectric signals tailored to the individual’s physiological and clinical characteristics. 

REAC technology has been successfully applied in various medical fields, including pain management [26,68,69], depression, anxiety, and stress-related disorders [34,36,37,70,71,72], neurological disorders [35,73,74,75,76], orthopedics [77,78,79,80], wound healing [30,81,82,83] anti senescence [84,85,86] and regenerative medicine [31,32,79,87,88,89]. Clinical trials have demonstrated the safety and tolerability of REAC technology, with no significant adverse effects reported.

### 2.12. REAC Treatments

REAC Neuro Postural Optimization (NPO) is a single-session non-invasive neurobiological modulation treatment [27,90] designed to improve postural control and stability through the modulation of the neurobiological system [27,28,46]. This therapy is based on the principle that the central nervous system (CNS) plays a critical role in maintaining postural stability and controlling movement [91]. By enhancing the function of the CNS, REAC NPO aims to optimize postural control and stability [38,73,75], which can help alleviate various musculoskeletal conditions, including chronic pain, muscle imbalances, and postural distortions, and improve the overall quality of life [90].

### 2.13. Neuro Psycho Physical Optimization 

Neuro Psycho Physical optimization (NPPO) is a non-invasive, neurobiological modulation treatment. NPPO has been shown to be effective in addressing a range of conditions and symptoms, including chronic pain, anxiety, depression, insomnia, fatigue [34,36,71,72,92,93,94], and cognitive impairment [33,74,95]. These benefits are achieved through the REAC functional optimization of the neurotransmission processes governed by endogenous bioelectrical activity. The NPPO is administered in a precise and fixed sequence of seven points on the ear [26,27,32,33,34,37,70,75].

In accordance with the treatment protocols, the NPPO treatment was administered for a duration of five consecutive days. During this period, it was permissible to administer the NPPO therapies at intervals of one hour, with a maximum of four therapies per day until a total of 18 therapies were completed.

### 2.14. Study Replicability

The replicability of the study is ensured by the REAC device’s fixed parameters for administering the NPO and NPPO treatments, which are determined by the manufacturer and cannot be altered by the operators. 

Additionally, the methods of administration have been thoroughly detailed in previous publications [27,28,34,71,93,95]. The study utilized devices REAC BENE 110 model (ASMED S.r.l, Florence, Italy).

### 2.15. Statistical Analysis

#### Statistics

For the statistical evaluation of the study, the GPower and SPSS 22 software was used.

In order to verify the normality of the data, we utilized the d’Agostino-Pearson test, which resulted in a DA-stat of 0.08 and a *p*-value of 0.95. Subsequently, we performed a paired *t*-test on the data, which demonstrated a highly significant *p*-value of *p* < 0.001, with a confidence interval of α = 0.05.

The study was conducted in accordance with the Declaration of Helsinki [96], and approved by the Ethics Committee of FEDERAL UNIVERSITY OF AMAPA—UNIFAP, MACAPÁ, Opinion number: 3.978.993. The trial was submitted for registration on the REBEC platform—Brazilian Registry of Clinical Trials—UTN 12 85 8902.

## 3. Results

Functional dysmetria was assessed immediately following the administration of NPO and at the conclusion of the 18-session NPPO treatments. For all other tests, follow-up occurred at the end of the 18-session NPPO program.

### 3.1. Functional Dysmetria Results

As previously demonstrated, the neurobiological treatment REAC NPO effectively induces the disappearance of functional dysmetria, as confirmed by the results of our study. We observed a consistent reduction of FD across all subjects, regardless of the severity of their initial condition.

Figure 1 provides a graphical summary of our findings, clearly showing how the REAC NPO treatment reset the previous DF values to zero, which remained stable throughout the follow-up period at the end of the 18 sessions of REAC NPO.

### 3.2. Timed Up and Go and Sit-To-Stand Results

Based on the results obtained from the TUG and STS tests, it can be observed that there was a significant difference (*p* < 0.001) between the pre and post-application of the REAC NPO and NPPO treatments. Table 1 represents the results of these tests.

These findings indicate that the application of REAC NPO and NPPO treatments had a statistically significant effect on the performance of the TUG test and sitting and rising from a chair in 30 s.

### 3.3. Fibromyalgia Impact Questionnaire Results

The results of the *t*-test revealed a statistically significant difference between the pre-and post-treatment scores (t = 3.25, *p* < 0.05), indicating that the intervention was effective in improving the participants’ fibromyalgia symptoms decreasing the impact of FM in participants QV Figure 2.

## 4. Discussion

Fibromyalgia is a chronic pain disorder that affects millions of people worldwide. The disorder is characterized by widespread musculoskeletal pain, fatigue, sleep disturbances, and cognitive difficulties [1]. While the exact cause of fibromyalgia is not yet fully understood, several factors have been suggested to contribute to its development [1,17,97].

One potential contributor to fibromyalgia is dysfunctional pain processing in the nervous system [2]. People with fibromyalgia appear to have an increased sensitivity to pain, and they may experience pain in response to stimuli that would not typically cause pain in others [2,98,99,100,101]. This sensitivity may be due to alterations in the way that pain signals are processed in the central nervous system. Studies have shown that individuals with fibromyalgia have changes in the way that their brains respond to pain stimuli [2,3,99,102,103,104,105], which may contribute to the heightened pain sensitivity and chronic pain experienced by these individuals.

In addition to changes in pain processing, fibromyalgia has also been associated with alterations in the hypothalamic-pituitary-adrenal (HPA) axis [106,107], which is a critical system involved in the body’s stress response [6,54]. Individuals with fibromyalgia often have lower levels of cortisol, a hormone that is released by the adrenal glands in response to stress [106,107,108]. This altered cortisol response may contribute to the fatigue and cognitive difficulties experienced by people with fibromyalgia [54,98,108].

Epigenetic factors may also play a role in the development of fibromyalgia [2,17,98]. Epigenetics refer to changes in gene expression that are not caused by alterations in the DNA sequence itself but instead by modifications to the DNA or the proteins that package it [109,110,111]. Research has shown that individuals with fibromyalgia may have alterations in DNA methylation, a process that can change the activity of genes [112,113,114]. These epigenetic changes may contribute to alterations in pain processing and stress response systems, as well as other aspects of fibromyalgia.

Finally, there may be behavioral and adaptive components to fibromyalgia [3,6,103,107,115]. For example, individuals with fibromyalgia may avoid physical activity due to fear of pain, which can lead to muscle weakness and further pain. Additionally, individuals with fibromyalgia may experience disrupted sleep, which can exacerbate pain and other symptoms [4,54,116]. These behavioral and adaptive factors may contribute to the chronicity of the disorder and make it challenging to manage.

The main finding of this study highlights the complex pathophysiological mechanisms that underlie fibromyalgia, including alterations in neurotransmission [4,6,22,98], perception modulation [2,100,104], environmental stress [6,54,107], and epigenetic modifications [17,110,112,113,114], all of which may affect the EBA [117,118,119,120,121,122,123,124,125,126,127]. 

Our study indicates that targeting the neurobiological features of the EBA through neurobiological modulation treatments could be a promising therapeutic strategy for individuals with fibromyalgia. Specifically, using REAC NPO and NPPO treatments to modulate the EBA may optimize both psychic and physical neurological responses to environmental factors, restoring proper function and potentially mitigating the symptoms of fibromyalgia, ultimately leading to an improved quality of life for patients.

Given that no other technologies have a similar mechanism of action to REAC technology, we can only evaluate the results of this study by comparing them to previous studies that utilized the same REAC treatment protocols in clinical scenarios where the pathophysiological factors are comparable to those of fibromyalgia.

Upon comparing the results, it can be concluded that the findings of this study align with those of previous studies that employed the same treatment protocols.

Although this study is an open-label study, which is a typical limitation, it is noteworthy that the limitations of this study are somewhat mitigated by the fact that the REAC NPO and NPPO treatments used in this study have already been extensively validated for their specific indications of use. Therefore, the established efficacy and safety of these treatments provide reassurance regarding their use in the current study.

## 5. Conclusions

Our study suggests that the use of safe and imperceptible non-invasive neurobiological modulation treatments, such as REAC NPO and NPPO, aimed at optimizing neuro-psycho-physical responses through a functional remodeling of the EBA, could be a promising therapeutic strategy for people with fibromyalgia.

Additionally, this study underscores the need for further investigation into the complex pathophysiology of fibromyalgia and the potential of neurobiological modulation treatments as a viable therapeutic option.

## Figures and Tables

**Figure 1 jpm-13-00902-f001:**
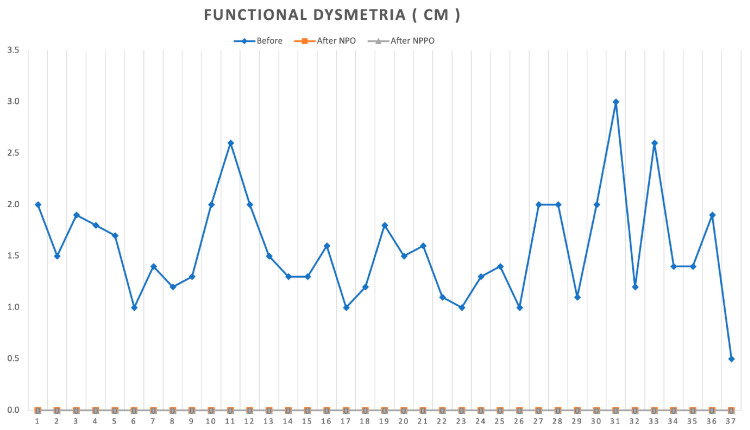
Graphical summary of the values of functional dysmetria before REAC NPO treatment (T0) and after treatment at T2 and T3.

**Figure 2 jpm-13-00902-f002:**
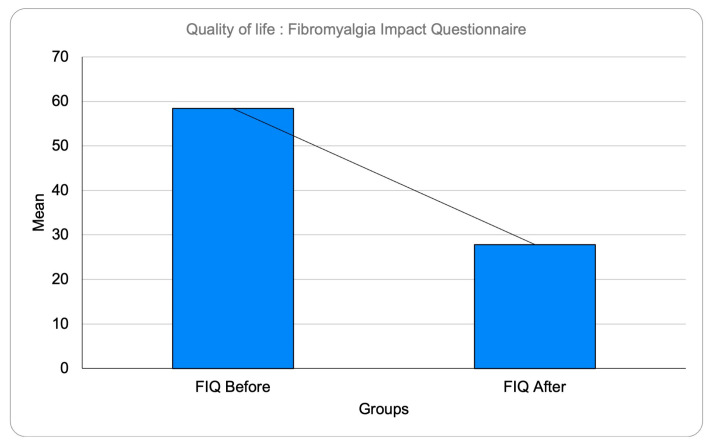
Significant decrease of FM’s impact on quality of life after therapy.

**Table 1 jpm-13-00902-t001:** TUG and STS test results.

	Mean ± Standard Deviation Before	Mean ± Standard Deviation After	*p*-Value
TUG (time)	12.53 ± 2.35	10.37 ± 1.67	≤0.001
STS tests	5.8 ± 1.27	7.5 ± 1.40	≤0.001

## Data Availability

The original data for this article is available at the following link for consultation: https://osf.io/n7kyt/, accessed on 25 May 2023.

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
