# Peer review of "REAC Neurobiological Modulation as a Precision Medicine Treatment for Fibromyalgia"

_jpm, 2023, doi:10.3390/jpm13060902_

Round 1
Reviewer 1 Report
Dear Authors
Thanks a lot for the opportunity you have offered me to revise the manuscript ""REAC Neurobiological Modulation as a Precision Medicine Treatment for Fibromyalgia "".
As a significant strength, this study evaluates the 17 effects of some REAC treatments on psychomotor responses and quality of life in 37 patients with 18 Fibromyalgia. This proposal is a novelty in the field and adds information to the existing evidence in the literature produced in the field.
As a major weakness, the manuscript sometimes needs more details and clarity concerning methodological steps that would help improve the understanding of the manuscript. Therefore, I have suggested some strategies to improve authors' reporting and increase the quality of their work.
Overall, my peer review is a major revision. I am available to revise the manuscript again after the amelioration.
#GENERAL:
*consider using active verbal forms instead of passive verbal forms. For example, ""this study aims…"" is better than ""The aim of this study was…""
* abbreviations: please, report in full all the abbreviations.
#ABSTRACT
* statistical values: please, report for the results their appropriate statistical values (e.g., p-value)
#INTRODUCTION
*background: there are several other theories underlying fibromyalgia that still need to be taken into account. For completeness, I invite the authors to read and integrate the following papers in their bibliographical references (doi: 10.1016/j.jbspin.2021.105153; doi: 10.1016/j.msksp.2022.102570; doi: 10.3389/fnhum.2020.00083)
*rationale: the rationale for the study needs to be better developed. Why should this study be done on fibromyalgia patient? This part needs to be better justified.
#METHODS.
*General. This study is presented as an ''open label interventional study''. I need help understanding what the authors mean. Is it a pilot study? Is it an rct? Is it a pre-post? I would ask them for more clarity on this point. They should also report the reporting guidelines for the study design used and adhere to them (e.g. Consort for rcts).
*power analysis: on which outcome was the sample size estimated? This is unclear. Also, if 20 subjects were sufficient, why did you study 37? The ""humanitarian reasons"" are unclear and not supported by the literature.
*population: 54%. Please, start the sentence with the word ""Fourty-five percent…"". Moreover, report mean and SD for the years.
*inclusion/exclusion criteria: they should be set using appropriate references.
*motor assessment: please, report the values for reliability and validity.
*ethics: please report the appropriate title for the ethical section. Moreover," the" Declaration of Helsinki ethical principles."" Please add the appropriate reference (doi: 10.1001/jama.2013.281053)
*statistical analysis: this section is completely missed. Please, add it. Moreover, delete the statistical info from the results section.
#RESULTS.
*NPPO program. You missed the point.
*data from the participants: please, report here the data from the participants and the statistical value to assess the baseline homogeneity. Moreover, add a flow chart of your study.
*General: please, delete the statistical info (e.g., type of analysis..) from the results section. Here you should only report the findings, the statistical value (p-value), and the value for dispersion (e.g., effect size) for each outcome.
#DISCUSSION.
*main findings: what are the main findings? Please, discuss them.
*General: you should discuss all the findings with appropriate pro and contrary literature.
*Limitations: this section is missed. Please, add it.
#CONCLUSION.
*implications for clinical practice and research: please, add this section.

Author Response
Dear Reviewer_1
Many thanks for your effort in reviewing our article "REAC Neurobiological Modulation as a Precision Medicine Treatment for Fibromyalgia" and especially for your suggestions to improve the quality and readability of the article.
#GENERAL:
*Consider using active verbal forms instead of passive verbal forms. For example, ""this study aims…"" is better than ""The aim of this study was…""
- Answer: Thanks for the linguistic suggestion.
* Abbreviations: please, report in full all the abbreviations.
- Answer: We double-checked in the text that the first abbreviation was preceded by the extended text
#ABSTRACT
* Statistical values: please, report for the results their appropriate statistical values (e.g., p-value)
#INTRODUCTION
*Background: there are several other theories underlying fibromyalgia that still need to be taken into account. For completeness, I invite the authors to read and integrate the following papers in their bibliographical references (doi: 10.1016/j.jbspin.2021.105153; doi: 10.1016/j.msksp.2022.102570; doi: 10.3389/fnhum.2020.00083)
- Answer: We have totally rewritten the introduction and as suggested inserted the three references in these sentences:
… Another factor that is probably common to CST and NIT in the genesis of fibromyalgia pain is the aberrant functioning of the nervous system, associated with small-fiber neuropathy [10.1016/j.jbspin.2021.105153].
and the second and third in this sentence:
Finally, PBF have also been implicated in the development and maintenance of fibromyalgia. Stress, depression, anxiety, and other psychological factors [10.1016/j.msksp.2022.102570, 10.3389/fnhum.2020.00083] can influence pain perception, sleep, and immune function, and may contribute to the onset and exacerbation of fibromyalgia symptoms.
*Rationale: the rationale for the study needs to be better developed. Why should this study be done on fibromyalgia patient? This part needs to be better justified.
- Answer: We have totally rewritten the rationale expanding it.
#METHODS.
*General. This study is presented as an ''open label interventional study''. I need help understanding what the authors mean. Is it a pilot study? Is it an rct? Is it a pre-post? I would ask them for more clarity on this point. They should also report the reporting guidelines for the study design used and adhere to them (e.g. Consort for rcts).
- Answer: we have improved the text as follows:
Study design - Open label interventional study, where both the researchers and the participants are aware of the treatment being administered, meaning there is no blinding or masking of the intervention. The study followed the ICH Guideline E8 (R1) on general considerations for clinical trials.
We have also included the study timeline paragraph:
Study timeline
T0 pre interventional assessments: functional dysmetria (FD), timed up and go test, sit-to-stand and fibromyalgia impact questionnaire; T1 administration of REAC NPO treatment; T2 verification of the effectiveness of the administration of the REAC NPO treatment by verifying the disappearance of the FD; T3 post interventional assessments: verification of the stability of the functional dysmetria correction, timed up and go test, sit-to-stand test and fibromyalgia impact questionnaire.
*Power analysis: on which outcome was the sample size estimated? This is unclear.
- Answer: Power analysis: was estimated for the results of the Fibromyalgia Impact Questionnaire.
Also, if 20 subjects were sufficient, why did you study 37? The ""humanitarian reasons"" are unclear and not supported by the literature.
- Answer: We enrolled all participants who expressed an interest in participating in the study, as they were from economically disadvantaged social classes. This study presented a valuable opportunity to potentially alleviate their suffering, and therefore, we included all individuals who had requested to participate.
*Population: 54%. Please, start the sentence with the word ""Fourty-five percent…"".
- Answer: Done
Moreover, report mean and SD for the years.
*Inclusion/exclusion criteria: they should be set using appropriate references.
- Answer: in the inclusion criteria we have rephrased as follows and included a reference:
Additionally, participants must not have engaged in physical exercise for at least three months, to avoid influences of training on physical performances, and must not have any musculoskeletal conditions that could interfere with their ability to participate in the evaluation [Selected Health Conditions and Likelihood of Improvement with Treatment].
*Motor assessment: please, report the values for reliability and validity.
- Answer: For the TUG we have insert this sentence:
The TUG performance is evaluated on a 5-point scale ranging from 1 (normal function) to 5 (severely abnormal function) based on the observer's perception of the individual's risk of falling[43].
For the STS we have insert this sentence:
In healthy adults, the time taken to complete a single STS repetition typically ranges from 2 to 3 seconds. However, this time can increase with age and in individuals with mobility impairments or other health conditions. The "five times sit-to-stand" test involves completing five repetitions of the STS task as quickly as possible, and normative values for this test range from approximately 11 to 15 seconds in healthy older adults.
For the functional dysmetria we have insert this sentence:
Functional dysmetria values do not exist, as it is a neurological diagnostic technique that focuses solely on the presence or absence of the phenomenon, rather than on quantitative measurements.
*Ethics: please report the appropriate title for the ethical section. Moreover," the" Declaration of Helsinki ethical principles."" Please add the appropriate reference (doi: 10.1001/jama.2013.281053)
- Answer: Done
*Statistical analysis: this section is completely missed. Please, add it. Moreover, delete the statistical info from the results section.
Answer: We have added the statistical analysis section and deleted the statistical info from the results section.
#RESULTS.
*NPPO program. You missed the point.
- Answer:
The schedule for administering the NPPO treatment has been included in the methods section, specifically within the paragraph dedicated to NPPO.
*Data from the participants: please, report here the data from the participants and the statistical value to assess the baseline homogeneity. ?????????????
Moreover, add a flow chart of your study.
- Answer: We have added the study timeline in the previous paragraphs
*General: please, delete the statistical info (e.g., type of analysis..) from the results section.
- Answer: Done
#DISCUSSION.
*Main findings: what are the main findings? Please, discuss them.
- Answer: To enhance the clarity and accessibility of our main findings, we have revised the previous text as follows:
The main finding of this study highlights the complex pathophysiological mechanisms that underlie fibromyalgia, including alterations in neurotransmission[4, 22, 96, 106], perception modulation[2, 98, 102], environmental stress[52, 105, 106], and epigenetic modifications[17, 109, 111-113], all of which may affect the EBA[116-126].
Our study indicates that targeting the neurobiological features of the EBA through neurobiological modulation treatments could be a promising therapeutic strategy for individuals with fibromyalgia. Specifically, using REAC NPO and NPPO treatments to modulate the EBA may optimize both psychic and physical neurological responses to environmental factors, restoring proper function and potentially mitigating the symptoms of fibromyalgia, ultimately leading to an improved quality of life for patients.
*General: you should discuss all the findings with appropriate pro and contrary literature.
- Answer: We have included the following text in the discussion section:
Given that no other technologies have a similar mechanism of action to REAC technology, we can only evaluate the results of this study by comparing them to previous studies that utilized the same REAC treatment protocols in clinical scenarios where the pathophysiological factors are comparable to those of fibromyalgia.
Upon comparing the results, it can be concluded that the findings of this study align with those of previous studies that employed the same treatment protocols.
*Limitations: this section is missed. Please, add it.
- Answer: To clarify the limitations of this study we have included the following text in the discussion section:
Although this study is an open-label study, which is a typical limitation, it is noteworthy that the limitations of this study are somewhat mitigated by the fact that the REAC NPO and NPPO treatments used in this study have already been extensively validated for their specific indications of use. Therefore, the established efficacy and safety of these treatments provide reassurance regarding their use in the current study.
#CONCLUSION.
*Implications for clinical practice and research: please, add this section.
- Answer: In order to provide a clearer understanding of the implications of our findings for both clinical practice and future research, we have included the following text in the added conclusion section:
Our study suggests that the use of safe and imperceptible non-invasive neurobiological modulation treatments, such as REAC NPO and NPPO, aimed at optimizing neuro-psycho-physical responses through a functional remodeling of the EBA, could be a promising therapeutic strategy for people with fibromyalgia.
Additionally, this study underscores the need for further investigation into the complex pathophysiology of fibromyalgia and the potential of neurobiological modulation treatments as a viable therapeutic option.

Reviewer 2 Report
The article entitled "REAC Neurobiological Modulation as a Precision Medicine Treatment for Fibromyalgia" is interesting and innovative. However, there are several points to be made.
Abstract:
Line 16: It should be written Radio Electric Asymmetric Conveyer (REAC).
Line 23: "the results showed" is not enough. Some numerical results should be given so that the reader has an idea of the magnitude of the benefit.
Introduction:
Line 43: Add a reference after (ABE).
Line 47: Add a reference after "fibromyalgia".
Line 48: Add a reference after (REAC).
Line 71: Remove "for humanitarian reasons".
Line 72: From which hospitals do patients come?
Line 79: It is unusual to identify a "brown" race.
Line 83: It is a bit contradictory to describe the selected population as "diverse" and to note that there is only one man for 36 women.
Line 89: In France the diagnosis of Fibromyalgia is more often made by Pain Doctors than by Rheumatologists.
Line 90: The criterion "must not have engaged in physical exercise" is unclear, as is "must not have any musculoskeletal conditions that could interfere". Please clarify.
Functional Dysmetria:
Line 115: Add a reference after "cerebellar dysfunction".
Line 117: Add a reference after "motor problems".
Study replicability:
Line 222: Add a reference after "publication".
Results:
Line 230: A brief description of the REAC procedure is needed. I suggest to show a picture of the device, to say that the system is connected to an earphone and to say the difference of stimulated territory in the ear between the NPO and NPPO procedure. It is also necessary to specify the duration of a stimulation session. Are NPO and NPPO performed in the same session? How long are the 18 sessions spread over? Five sessions per week?
Line 241: The legend of the figure is not well understood. Does the value in cm correspond to the difference after the first session or after the 18 sessions?
Author Response
Dear Reviewer_2
Many thanks for your effort in reviewing our article "REAC Neurobiological Modulation as a Precision Medicine Treatment for Fibromyalgia" and especially for your suggestions to improve the quality and readability of the article
Abstract:
Line 16: It should be written Radio Electric Asymmetric Conveyer (REAC).
- Answer: Done
Line 23: "the results showed" is not enough. Some numerical results should be given so that the reader has an idea of the magnitude of the benefit.
Introduction:
Line 43: Add a reference after (ABE).
- Answer: the references had already been entered [11, 12]
Line 47: Add a reference after "fibromyalgia".
Line 48: Add a reference after (REAC).
- Answer: Thanks for this suggestion, but to avoid repetitions we preferred to insert the references at the end of the sentence.
Line 71: Remove "for humanitarian reasons".
- Answer: Done
Line 72: From which hospitals do patients come?
- Answer: As stated in our study, participants were recruited from diverse sources, rather than a single hospital. (The study recruited a total of 37 fibromyalgic participants using a non-probabilistic sampling method that included written and electronic dissemination of information, as well as personal outreach to associations of patients with FS and rheumatologic clinics.)
Line 79: It is unusual to identify a "brown" race.
- Answer: In the region of the Amazon where the study was conducted, the term "brown" is commonly used and understood. This term refers to the self-declaration of skin color ,provided for in the Statute of Racial Equality, object of Brazilian Law No. 12,288, of 2010,
Line 83: It is a bit contradictory to describe the selected population as "diverse" and to note that there is only one man for 36 women.
- Answer: Thanks for this observation. We have described the selected population as "different" not by gender but by different racial typology, and specially to the varying degrees of FS severity and treatment histories ( line 131 ). In this sense, "diversity" did not seem contradictory to us.
Line 89: In France the diagnosis of Fibromyalgia is more often made by Pain Doctors than by Rheumatologists.
- Answer: Thanks for this clarification. In Brazil and Italy, the diagnosis of fibromyalgia is often made by a rheumatologist.
Line 90: The criterion "must not have engaged in physical exercise" is unclear, as is "must not have any musculoskeletal conditions that could interfere". Please clarify.
- Answer: In the sentence we added
Additionally, participants must not have engaged in physical exercise for at least three months, “to avoid influences of training on physical performances”,
and to clarify
“must not have any musculoskeletal conditions that could interfere with their ability to participate in the evaluation” we have added a reference.
Functional Dysmetria:
Line 115: Add a reference after "cerebellar dysfunction".
- Answer: we added the reference (Ataullah, A. H. M. and I. A. Naqvi. "Cerebellar dysfunction." In Statpearls. Treasure Island (FL): StatPearls Publishing Copyright © 2023, StatPearls Publishing LLC., 2023)
Line 117: Add a reference after "motor problems".
- Answer: we added the reference (Rasouli, O., E. A. Fors, P. C. Borchgrevink, F. Ohberg and A. K. Stensdotter. "Gross and fine motor function in fibromyalgia and chronic fatigue syndrome." J Pain Res 10 (2017): 303-09. 10.2147/JPR.S127038)
Study replicability:
Line 222: Add a reference after "publication".
- Answer: as requested we have inserted the appropriate citations
Results:
Line 230: A brief description of the REAC procedure is needed. I suggest to show a picture of the device, to say that the system is connected to an earphone and to say the difference of stimulated territory in the ear between the NPO and NPPO procedure. It is also necessary to specify the duration of a stimulation session. Are NPO and NPPO performed in the same session? How long are the 18 sessions spread over? Five sessions per week?
- Answer: To avoid self-plagiarism, we have chosen not to repeat the administration procedures of the two treatments in this manuscript. However, we have taken your feedback into consideration and have included appropriate references to previous manuscripts where the administration procedures are described in detail.
We have also included a study timeline to provide a clearer understanding of the timelines that were involved in the study.
Line 241: The legend of the figure is not well understood. Does the value in cm correspond to the difference after the first session or after the 18 sessions?
- Answer: We have revised the figure caption as follows:
Figure 1. Graphical summary of the values of functional dysmetria before REAC NPO treatment (T0) and after treatment at T2 and T3.

Round 2
Reviewer 1 Report
Dear authors,
Thank you for your revision which has greatly improved the introduction and discussions.
Regarding the materials and methods, there are still some weak points:
- the explication of the variable used to calculate the sample size in the main text is missing. Please add it.
- it is not clear to me why the authors used "the ICH Guideline E8 (R1) on general considerations 108 for clinical trials[40]" instead of CONSORT (internationally recognised guideline). Furthermore, the link in ref no. 40 does not work. Please adopt CONSORT for your reporting.
- "Open label interventional study" continues to be mysterious to me. In fact, if the study aims to investigate efficacy, the design is a randomised controlled trial (which can also be open label), not anything else. Please be clearer.
-As an RCT, prospective registration in international databases is mandatory. This point is completely missing. Please state whether you have done so or not.
I ask the authors to resolve these missing points so that we can assess the methodological robustness of their work.
Best regards.
Author Response
Jpm-2306142 REAC Neurobiological Modulation as a Precision Medicine Treatment for Fibromyalgia
Point to point answers (Round 2)
Dear Revisor 1,
Thank you very much for your considerations, which will certainly help to further refine the work. Here are some clarifications and considerations.
Regarding the materials and methods, there are still some weak points:
- the explication of the variable used to calculate the sample size in the main text is missing. Please add it.
Answer: Indeed, the calculation presented pertains to the sample size power analysis discussed in lines 112-115, which indicates that a sample of 20 individuals would be sufficient for a general population. However, the inclusion of the additional 15 individuals was a matter of convenience for both the researchers and the individuals themselves, who expressed their willingness to volunteer for the study. From a scientific standpoint, we deem this information unnecessary.
- it is not clear to me why the authors used "the ICH Guideline E8 (R1) on general considerations 108 for clinical trials [40]" instead of CONSORT (internationally recognised guideline). Furthermore, the link in ref no. 40 does not work. Please adopt CONSORT for your reporting.
- "Open label interventional study" continues to be mysterious to me. In fact, if the study aims to investigate efficacy, the design is a randomised controlled trial (which can also be open label), not anything else. Please be clearer.
- Answer: To clarify the mystery, below is the explanation of an "open-label interventional study"
An open-label interventional study is a type of clinical research in which both the investigator and the participants know which treatment is being given. In this type of study, the researchers do not use a placebo or a control group. Instead, all participants receive the treatment being tested.
Interventional studies are those where the researchers actively intervene in the study population by administering a treatment or intervention. Open-label studies are different from blinded studies, in which the participants do not know whether they are receiving the treatment or a placebo.
Open-label studies are often used in early-stage clinical trials, where the primary goal is to assess the safety and feasibility of a treatment. In some cases, open-label studies may also be used in later-stage clinical trials, particularly if blinding is not feasible due to the nature of the intervention or the outcome being measured.
Overall, open-label interventional studies have some limitations as they are vulnerable to bias, both from participants and researchers, and they may not provide strong evidence of treatment efficacy. However, they can be useful in exploring new treatments and providing preliminary data for future studies.
(Interventional studies can be divided broadly into two main types: (i) “controlled clinical trials” (or simply “clinical trials” or “trials”), in which individuals are assigned to one of two or more competing interventions https://www.cancer.gov/publications/dictionaries/cancer-terms/def/open-label-study
In interventional studies, the researcher actively interferes with nature – by performing an intervention in some or all study participants – to determine the effect of exposure to the intervention on the natural course of events.
https://www.ncbi.nlm.nih.gov/pmc/articles/PMC6647894/#:~:text=Interventional%20studies%20can%20be%20divided,entire%20groups%2C%20e.g.%2C%20villages%2C
The current study was also titled as an open-label trial due to the absence of randomization, blinding procedures, and a control group, so that it is not randomized nor controlled. The CONSORT statement in turn provides guidelines for randomized clinical trials, including the recommendation to mention randomization in the title of the study. As this was an initial study without those characteristics of a typical RCT, we have respectfully opted out of using the CONSORT statement. This is also clear to us a limitation of the study.
-As an RCT, prospective registration in international databases is mandatory. This point is completely missing. Please state whether you have done so or not.
- Answer: The trial was submitted for registration on the REBEC platform https://ensaiosclinicos.gov.br/ UTN 12 85 8902, but we are still awaiting approval, so we chose to omit and mention only the approval by the opinion of the local ethics registry.
We trust that we have duly considered and clarified all the points.
Kind regards
Reviewer 2 Report
The article entitled "REAC neurobiological modulation as a precision medicine treatment for fibromyalgia" has improved with a very good introduction on the physiopathology of fibromyalgia. On the other hand, the section devoted to REAC technology remains cluttered and uninformative. However, there are at least 2 very good articles in the literature that simply explain a rather complex theory: 1) Machado VG, Brun ABS, Manffra EF. Effects of the radio electric asymmetric conveyer (REAC) on motor disorders: An integrative review. Front Med Technol. 2023 Feb 27;5:1122245. doi: 10.3389/fmedt.2023.1122245. PMID: 36923595; PMCID: PMC10009233. 2) Rinaldi S, Mura M, Castagna A, Fontani V. Long-lasting changes in brain activation induced by a single REAC technology pulse in Wi-Fi bands. Randomized double-blind fMRI qualitative study. Sci Rep. 2014 Jul 11;4:5668. doi: 10.1038/srep05668. PMID: 25011544; PMCID: PMC4092330.
Materials and Methods :
Study time line: what is the spacing between T0, T1, T2 and T3?
Line 136 : most work. What is the % ?
Line 148: What are the "musculoskeletal conditions"? The reference 41 to which the authors refer is incomplete. In particular, the publisher and possibly the pages corresponding to the item discussed are missing.
In the chapter "Methods" I suggest to better classify the subchapters by putting in order 1) The treatment by REAC technology, detailing the technique. How many NPO sessions and how many NPPO sessions, specifying the duration of the sessions and the total duration of the sessions. The stimulation parameters must also be detailed. For example for the NPO protocol: a single burst with a duration of 250 ms and a frequency in the train of 5.8 GHz. The same thing for the NPPO protocol. With the information available in this article, it is impossible that someone could reproduce this protocol, which is still the purpose of an article. We would need a picture that shows the pinna being stimulated and the exact area depending on the protocol. 2) The different evaluation methods.
Figure 1: It is difficult to understand this figure. Moreover, the " functional dysmetria " having been evaluated after the NPO procedure and the 18 sessions of NPPO, what are the values represented in the figure?
Table 1: Standard deviations or deviations from the mean should be added.
Figure 2: Standard deviations or deviations from the mean should be added.
Author Response
Jpm-2306142 REAC Neurobiological Modulation as a Precision Medicine Treatment for Fibromyalgia
Point to point answers (Round 2)
Dear Reviewer 2,
Thank you very much for your considerations, which will certainly help to further refine the work . Here are some clarifications and considerations
The article entitled "REAC neurobiological modulation as a precision medicine treatment for fibromyalgia" has improved with a very good introduction on the physiopathology of fibromyalgia. On the other hand, the section devoted to REAC technology remains cluttered and uninformative. However, there are at least 2 very good articles in the literature that simply explain a rather complex theory:
1) Machado VG, Brun ABS, Manffra EF. Effects of the radio electric asymmetric conveyer (REAC) on motor disorders: An integrative review. Front Med Technol. 2023 Feb 27;5:1122245. doi: 10.3389/fmedt.2023.1122245. PMID: 36923595; PMCID: PMC10009233.
2) Rinaldi S, Mura M, Castagna A, Fontani V. Long-lasting changes in brain activation induced by a single REAC technology pulse in Wi-Fi bands. Randomized double-blind fMRI qualitative study. Sci Rep. 2014 Jul 11;4:5668. doi: 10.1038/srep05668. PMID: 25011544; PMCID: PMC4092330.
- Answer: Machado VG, Brun ABS, Manffra EF. Effects of the radio electric asymmetric conveyer (REAC) on motor disorders: An integrative review. Front Med Technol. 2023 Feb 27;5:1122245. doi: 10.3389/fmedt.2023.1122245. PMID: 36923595; PMCID: PMC10009233.
This reference has been added as [90]
- Answer: Rinaldi S, Mura M, Castagna A, Fontani V. Long-lasting changes in brain activation induced by a single REAC technology pulse in Wi-Fi bands. Randomized double-blind fMRI qualitative study. Sci Rep. 2014 Jul 11;4:5668. doi: 10.1038/srep05668. PMID: 25011544; PMCID: PMC4092330.
This reference was already cited as [26] now [27]
Materials and Methods:
Study time line: what is the spacing between T0, T1, T2 and T3?
- Answer: Added to lines 122 and 123
Line 136 : most work. What is the % ?
- Answer: Thanks for the observation, it was included in the section (line 139).
Line 148: What are the "musculoskeletal conditions"? The reference 41 to which the authors refer is incomplete. In particular, the publisher and possibly the pages corresponding to the item discussed are missing.
- Answer: We included examples of musculoskeletal conditions that could hinder the performance of the tests and corrected the reference ( lines 151-153)
The treatment by REAC technology, detailing the technique. How many NPO sessions and how many NPPO sessions, specifying the duration of the sessions and the total duration of the sessions.
- Answer: The application technique has been comprehensively documented in the extensive literature cited, and it is worth noting that additional information has been provided on lines 295. Regarding the NPO single session, specific emphasis has been placed on lines 121 and 284 to underscore the importance of this information. The comprehensive description of the number of NPPO sessions, their duration, and the total number of sessions has already been elucidated in lines 295-300, as well as in the section addressing study replicability (lines 303-309). Moreover, the thorough description of the mechanisms underlying the REAC technology (lines 256-300) firmly establishes the absence of any stimulatory character in the technique.
The stimulation parameters must also be detailed. For example for the NPO protocol: a single burst with a duration of 250 ms and a frequency in the train of 5.8 GHz. The same thing for the NPPO protocol. With the information available in this article, it is impossible that someone could reproduce this protocol, which is still the purpose of an article. We would need a picture that shows the pinna being stimulated and the exact area depending on the protocol. 2) The different evaluation methods.
- Answer: We confirm that the purpose of scientific publications is to provide the scientific community with the opportunity to learn about new methodologies. In this case, as in others, not only the parameters but also the peculiarities with which the parameters are generated are essential data. Therefore, specifying "a single burst with a duration of 250 ms and a frequency in the train of 5.8 GHz" without providing the specific technology that generates them does not allow for reproducibility. For this reason, in the manuscript, we have created a section on Study Replicability in the Materials and Methods. If the reviewer is interested in reproducing the study, we would be happy to provide both the equipment we used and the training necessary to use it.
Figure 1: It is difficult to understand this figure. Moreover, the " functional dysmetria " having been evaluated after the NPO procedure and the 18 sessions of NPPO, what are the values represented in the figure?
- Answer: Figure 1 has been redone for better understanding.
Table 1: Standard deviations or deviations from the mean should be added.
- Answer: Done
Figure 2: Standard deviations or deviations from the mean should be added.
- Done
We trust that we have duly considered and clarified all the points.
Best regards